# Critical Stress Determination of Local and Distortional Buckling of Lipped Angle Columns under Axial Compression

**Junfeng Zhang** [1], **Bo Li** [1], **Anqi Li** [2,*] **and Shiyun Pang** [3]

[1] School of Civil Engineering, Zhengzhou University, Zhengzhou 450001, China; zhangjunfeng@zzu.edu.cn (J.Z.); 202022212114199@gs.zzu.edu.cn (B.L.)
[2] School of Civil Engineering, Shandong Jianzhu University, Jinan 250101, China
[3] School of Civil Engineering, Chongqing University, Chongqing 400045, China; shiyun0820@163.com
[*] Correspondence: anqi666999@126.com

**Abstract:** In recent years, cold-formed steel has been widely used in prefabricated steel structures, and the common cross-section forms are mainly complex lipped angle sections. However, there is a lack of design guidance for such a cross-section due to the complex geometric property. The restraint between adjacent plates cannot be considered proper for the traditional analytical method. Therefore, it is particularly important to study the stability bearing capacity of angle sections with complex edges under axial compression. In this paper, the finite strip software (CUFSM5) was used to analyze the critical stress of 1296 different angle sections under axial compression. The deformation diagram and the critical stress of elastic buckling were obtained. Considering the restraint between adjacent plates, the formula for predicting the critical stress of elastic local buckling of complex lipped angle sections was proposed and verified. Further, the critical stress of elastic distortional buckling of 918 complex lipped angle sections was analyzed by CUFSM. It was found that the cross-sections can be divided into two categories: cross-section without distortional point and cross-section with distortional point. It was found that the critical stress of elastic local buckling of the angle steel section can be significantly improved by the complex edge. Additionally, the critical stress of elastic local buckling of the section is less affected by the edge size for the complex edge section. The accuracy of the Hancock method for calculating the critical stress of elastic distortional buckling of complex lipped angle sections with distortional points was verified. The presented research can provide useful guidelines for designing cold-formed steel angle columns.

**Keywords:** cold-formed steel; angle column; complex edge; finite strip method; critical stress of elastic buckling

## 1. Introduction

In recent years, with the rapid development of prefabricated construction, steel structure has entered a new development stage. It has been widely used in construction sites, shops, offices, power supply facilities, and temporary buildings [1–3]. In the critical period of coping with the epidemic prevention and control of COVID-19, prefabricated steel structure movable house plays a vital role [4,5]. In prefabricated buildings, the wall thickness of cold-formed steel members should not be greater than 6 mm or less than 1.5 mm, and the wall thickness of main load-bearing structural members should not be less than 2 mm [6]. The buckling modes of cold-formed steel members can mainly be divided into global buckling, local buckling, and distortional buckling. The design methods of cold-formed thin-walled steel members mainly include "the effective width method" and "the direct strength method" (DSM) [7].

The guiding idea of the effective width method is to consider the influence of the post buckling strength of the plate. The yield strength of the steel was not changed, and the effective section of the plate was obtained by reducing the width of the plate. In this

way, the bearing capacity of the component can be obtained based on the effective section. Although the calculation of the direct strength method is greatly simplified compared with the effective section method, only the single buckling mode can be considered, neglecting the correlation among the different buckling modes [7]. In addition, the direct strength method is proposed for common cross-sections, such as C-shaped and Z-shaped cross-sections, as shown in Figure 1.

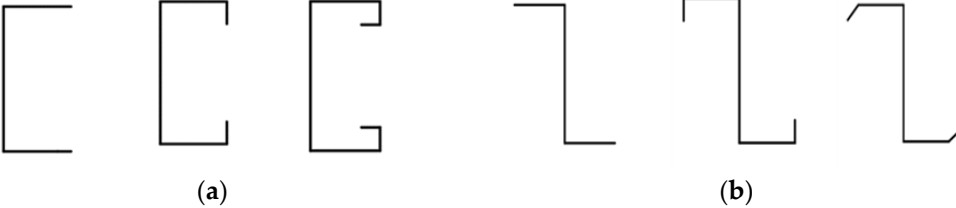

(a)                                                                                        (b)

**Figure 1.** Common section forms of cold-formed steel members: (**a**) C-shaped cross-sections; (**b**) Z-shaped cross-sections.

However, to meet the needs of waterproof, thermal insulation, heat insulation, sound insulation, and other architectural functions, some changes to the cross-section may be necessary, such as special complex lipped angle members, as shown in Figure 2. The x0 and y0 are the centroidal axes, and the x and y are the centroidal principal axes of the section, where a is the length of the long limb and b is the length of the short limb, while c and d are the length of the primary lip and the secondary lip, respectively. However, the design theory and experimental research regarding this type of section are limited. Therefore, it is particularly important to explore the mechanical property and design method of the cold-formed steel members with complex edges.

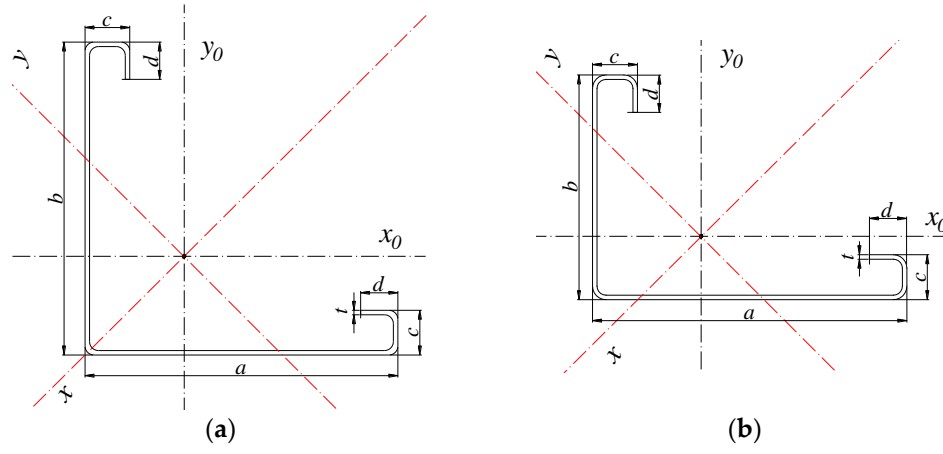

(a)                                                                                        (b)

**Figure 2.** Complex lipped angle member: (**a**) complex lipped angle steel with equal limb; (**b**) complex lipped angle steel with unequal limb.

Research on elastic critical stress and stable bearing capacity of different cold-formed steel sections has been rising in recent years. Rasmussen [8] predicted the bearing capacity of cold-formed equal-limb angular cross-section columns using the DSM. Young and Ellobody [9] analyzed the test results of unequal limb angle steel columns under axial compression. It was found that only local and global buckling was observed, and there was no distortional buckling during the test. In addition, Young and Chen [10] conducted in-depth research on the mechanical behavior of the non-symmetric cold-formed steel angular column, and it was reported that the North American Specification (AISI S100) [11] for the design of cold-formed steel non-symmetric lipped angular columns are generally quite conservative. Santos et al. [12] determined the criterion for judging the buckling modes of lipped-edge groove steel members under compression through simulation. The structural strength and stability of cold-formed steel lipped channel beam columns under

biaxial moments and axial forces were experimentally investigated by Torabian et al. [13]. Zhang et al. [14] conducted the vertical bearing capacity test of the prefabricated box houses, and the calculation method for the initial rotation stiffness was obtained. Then Zhang et al. [15] carried out the test on the full-scale models of prefabricated box houses under longitudinal horizontal load. The test results showed that the lateral stiffness of the structure was small and lateral support should be set. Zhang et al. [16] conducted the test and finite element simulation analysis of the flexural stiffness of the bottom of a prefabricated box house, and the results showed that the finite element simulation results were in good agreement with the experimental results and the reliability of the finite element analysis was verified. Park et al. [17] proposed an embedded steel column-to-foundation connection for modular structural systems and evaluated the behavior of the proposed connection through experimental research and finite element analysis (FEA).

With the rapid development of prefabricated buildings, it is urgent to study the stability behavior of cold-formed steel members under axial compression, especially the cold-formed steel members with complex edges. Accordingly, the direct strength method has been paid more attention in recent years. It is necessary to solve the critical stress of elastic buckling for the direct strength method. Therefore, the stress of elastic buckling is the first problem in obtaining the stability bearing capacity of the cold-formed steel members under axial compression. However, there is no guideline for determining the stress of elastic buckling of the complex lipped angle steel members.

In this paper, the finite strip software CUFSM was used to analyze the critical stress of elastic local buckling of different sections under axial compression, and the calculation formulas of critical stress of elastic buckling of lipped angle sections were proposed by regression analysis. Further, the critical stress of elastic distortional buckling of complex lipped angle sections was analyzed by CUFSM [18]. The accuracy of the Hancock [19] method for calculating the critical stress of elastic distortional buckling of complex lipped angle sections with distortional points was verified. The present research work is beneficial for the design and optimization of the cold-formed steel members with complex edges. However, it should be mentioned that the present method is only applicable to cold-formed thin-walled members under axial compression.

## 2. CUFSM for Solving Critical Stress of Elastic Buckling

### 2.1. Introduction of CUFSM

The numerical analysis method has been proven reliable and has always been the most commonly used method for solving elastic buckling stress. CUFSM is one of the most commonly used finite strip software to solve the critical stress of elastic buckling [20]. The main process of solving the critical stress of elastic buckling by CUFSM included: (1) creating sections; (2) dividing units; (3) imposing constraints and boundary conditions; (4) loading; (5) analytical solution.

Take the section of complex lipped angle steel with equal limbs shown in Figure 2a as an example. It should be mentioned that common cross-sections such as C- and Z-shaped can be created directly by inputting the cross-section dimensions. For the novel cross-section form with a complex edge, it is impossible to directly generate the cross-section by inputting cross-section dimensions. It is necessary to create the cross-section by coordinates of some key points. The division of the cross-section and the deformation diagram of the complex lipped angle steel with equal limbs are shown in Figure 3.

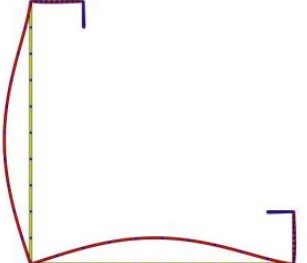

**Figure 3.** The deformation diagram of the complex lipped angle steel with an equal limb.

When the pressure at all nodes is set to be 1 kN, the typical analysis result is the diagram of the relationship between half wavelength and stress, as shown in Figure 4.

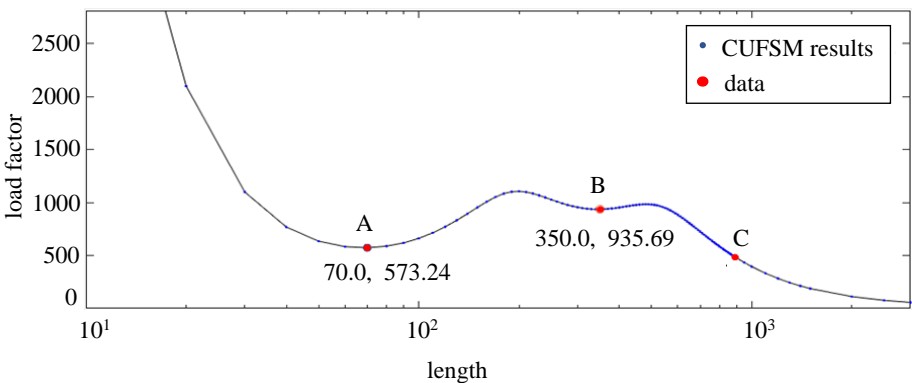

**Figure 4.** CUFSM analysis curve.

As shown in Figure 4, the ordinate of the first extreme point A is the critical stress of elastic local buckling, which is 573.24 MPa. The abscissa of point A indicates that the half wavelength of local buckling is 70 mm. The second minimum point suggests that the critical stress of elastic distortional buckling of the section is 935.69 MPa, and the half wavelength of distortional buckling of the section is 350 mm. Only the final descent section is taken as the critical stress of the elastic global buckling of the member.

### 2.2. Verification of CUFSM

To verify the accuracy of CUFSM modeling, the example in Direct Strength Method (DSM) Design Guide 2006 was used [21]. The section size of the example is shown in Figure 5. The critical stress and buckling behaviors were compared between the design guide and CUFSM, as shown in Figure 6.

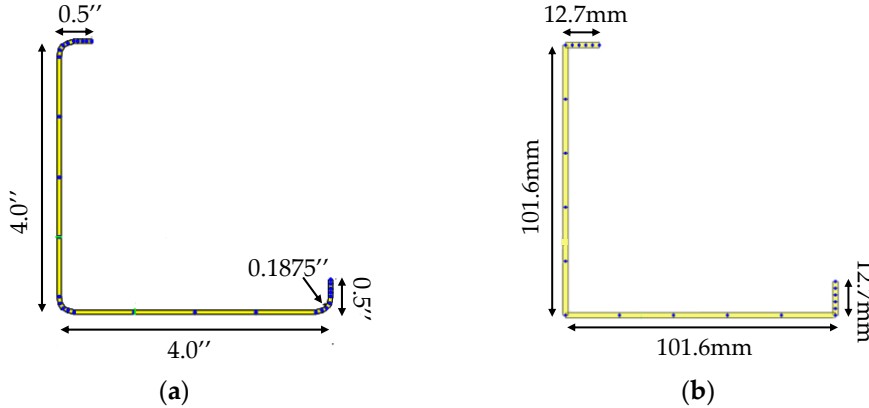

**Figure 5.** Sectional parameters. (**a**) American units; (**b**) Chinese units.

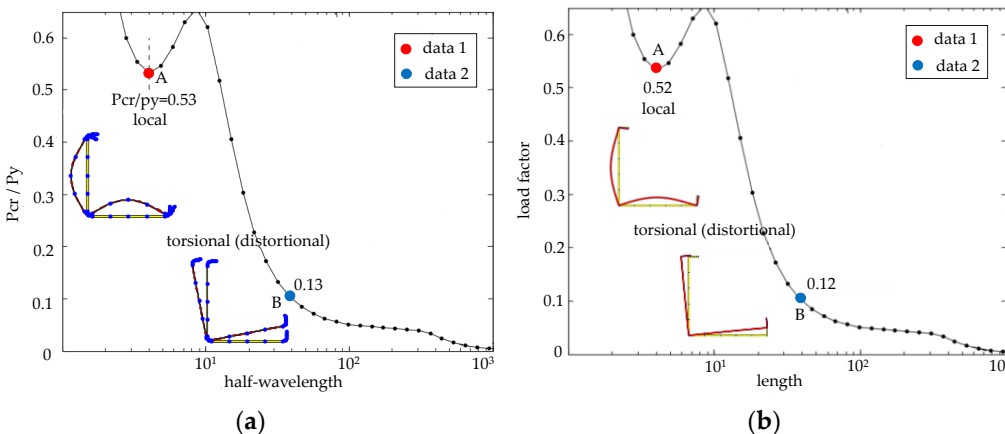

**Figure 6.** Buckling behaviors. (**a**) The deformation diagram of the design guide; (**b**) The deformation diagram of CUFSM.

From a comparison of diagram and critical stress between the design guide and the CUFSM in this paper, it can be seen that the analysis results of CUFSM are the same as those of the design guide. In this way, the accuracy of the CUFSM modeling can be validated.

*2.3. The Development of MATLAB Version of CUFSM*

In this paper, MATLAB was adopted to provide a more flexible application of CUFSM. The CUFSM was called by MATLAB programming, and the analysis results were processed by MATLAB to obtain the corresponding critical stress of elastic local and distortional buckling.

Only section sizes were needed to be put in the MATLAB version of CUFSM. As shown in Figure 7, the section number is the input sequence of the section. *Li* denotes the length of the member, and the other geometric parameters were defined in Figure 2. *LB* represents local buckling. $\lambda$ represents a half wavelength of local buckling, and $P_{cr}$ is the critical stress of elastic local buckling.

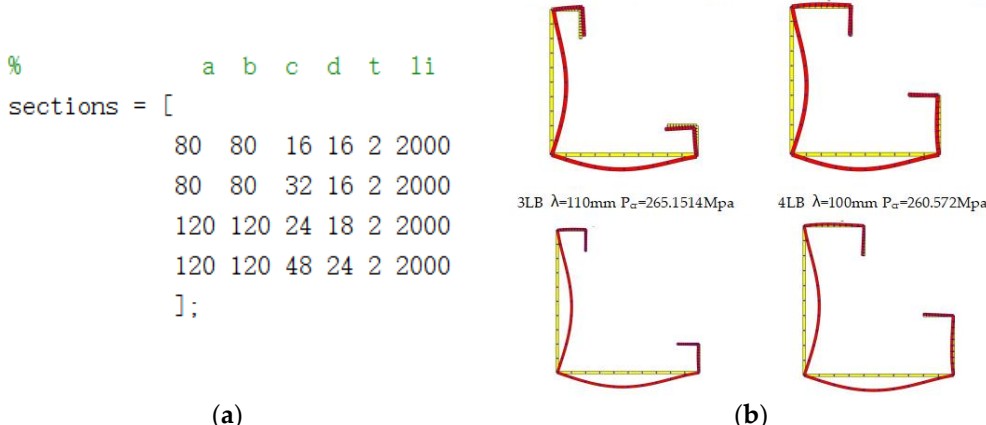

**Figure 7.** Critical stress analysis of elastic local buckling: (**a**) Input of section size; (**b**) Analysis result.

## 3. Critical Stress of Elastic Local Buckling of Angle Columns under Axial Compression

*3.1. Sectional Dimension*

The strength grade of steel used in this paper is Q345 [6], with a nominal yield strength of 345 MPa. To ensure the applicability of the formulas, according to the technical code for cold-formed thin-walled steel structures of China [6] and North American Specification (AISI S100) [11], the range of sectional dimensions of each plate of the complex lipped angle sections is determined, as listed in Table 1. The definition of specific parameters is shown in Figure 2.

**Table 1.** Parameters of the complex edge.

| Series | *a/t* | *a/b* | *c/a* | *d/a* | *d/c* |
|---|---|---|---|---|---|
| Complex edge | 40~200 | 1.0~1.5 | 0.2~0.5 | 0.1~0.3 | 0.5~1.0 |

Note: *a/t* is the ratio of long limb to thickness; *a/b* is the ratio of long limb to short limb; *c/a* is the ratio of primary lip to long limb; *d/a* is the ratio of secondary lip to long limb; *d/c* is the ratio of secondary lip to primary lip.

According to the parameters in Table 1, various sections can be selected, and the plate thickness is determined as 2 mm. The length to thickness ratio was 40, 60, 80, 100, 120, 160, 180, and 200, respectively. The primary lip to long limb ratio was 0.2, 0.25, 0.30, 0.35, 0.4, 0.45 and 0.5, respectively. The secondary lip to primary lip ratio was 0.5, 0.75, and 1, respectively. The long limb to short limb ratio was 1.1, 1.2, 1.3, 1.4, and 1.5 for the unequal limb angle section, respectively.

According to the above parameters, a total of 1296 different types of angle sections were designed, including 63 simple lipped equal limb sections, 315 simple lipped unequal limb sections, 153 complex lipped equal limb sections and 765 complex lipped unequal limb sections. The critical stresses of elastic local buckling of all sections are obtained using the MATLAB version of CUFSM. The calculation formulas for critical stress of elastic local buckling of lipped angle steel sections were obtained by regression analysis considering the constraint between adjacent plates, as shown in Figure 8. $k_l$ is the effect of the lip on the critical stress of angle steel section, and $k_f$ is the effect of limb length ratio on critical stress of angle steel section.

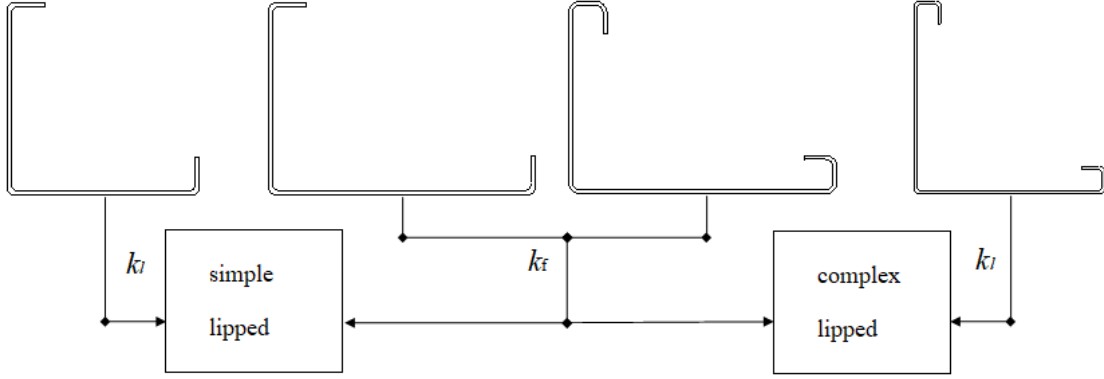

**Figure 8.** Critical stress analysis of elastic local buckling.

*3.2. Critical Stress of Elastic Local Buckling of Simple Lipped Angle Members*

For the section of simple lipped equal limb angle steel, only the effect of the lip on the critical stresses of elastic local buckling should be considered. Table 2 shows the critical stress of elastic local buckling of the 63 simple lipped equal limb angle steel sections.

**Table 2.** Critical stresses of elastic local buckling for simple lipped equal limb angle member (unit: MPa).

| *a/t* | Critical Stress of Elastic Local Buckling with Different *c/a* | | | | | | |
|---|---|---|---|---|---|---|---|
| | 0.15 | 0.20 | 0.25 | 0.30 | 0.35 | 0.40 | 0.45 |
| 40 | 515.63 | 518.02 | 512.94 | 499.99 | 472.89 | 433.36 | 384.09 |
| 60 | 232.11 | 231.96 | 229.06 | 222.53 | 210.65 | 192.60 | 170.86 |
| 80 | 131.13 | 130.81 | 129.05 | 125.25 | 118.57 | 108.40 | 96.14 |
| 100 | 84.09 | 83.82 | 82.66 | 80.20 | 75.89 | 69.40 | 61.54 |
| 120 | 58.46 | 58.25 | 57.42 | 55.71 | 52.72 | 48.19 | 42.74 |
| 140 | 42.97 | 42.80 | 42.19 | 40.94 | 38.73 | 35.40 | 31.40 |
| 160 | 32.91 | 32.78 | 32.30 | 31.34 | 29.65 | 27.11 | 24.04 |
| 180 | 26.01 | 25.90 | 25.53 | 24.76 | 23.43 | 21.42 | 19.00 |
| 200 | 21.07 | 20.98 | 20.68 | 20.06 | 18.98 | 17.35 | 15.39 |

The relationship between the critical stress of elastic local buckling of simple lipped equal limb angle section and the ratio $c/a$ is shown in Figure 9. It can be seen that the critical stresses of elastic local buckling are nonlinear with $c/a$ as the ratio of length to thickness ($a/t$) varies. When the length of the section is constant, the critical stresses of elastic local buckling of the section decrease with the increase of $c/a$. The main reason is that the simple lip belongs to a partially restrained plate, and the deformation of the lip increases with the increase of its size, which weakens the section's ability to resist local buckling of the plate. To further determine the effect of $c/a$ on the critical stress of elastic local buckling, the expression of $k_l$ is determined by the relationship curve of $c/a$ and $k_l$.

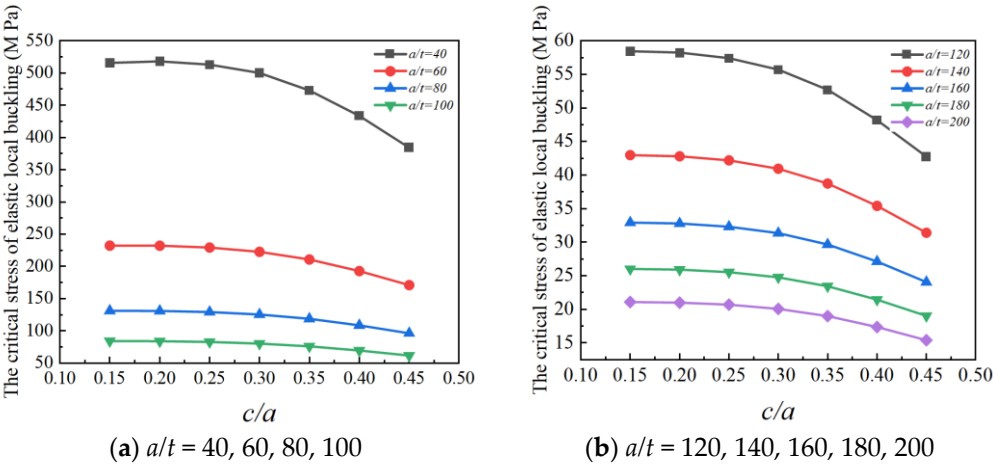

(**a**) $a/t$ = 40, 60, 80, 100  (**b**) $a/t$ = 120, 140, 160, 180, 200

**Figure 9.** Relationship between critical stress of elastic local buckling and $c/a$.

In addition, when $a/t$ = 40, the buckling behavior is shown in the following Figure 10:

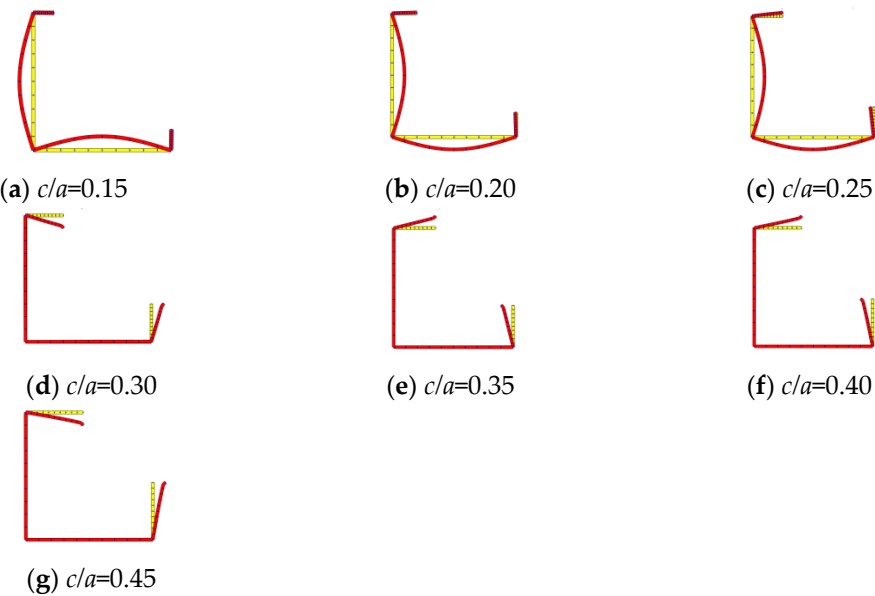

(**a**) $c/a$=0.15  (**b**) $c/a$=0.20  (**c**) $c/a$=0.25

(**d**) $c/a$=0.30  (**e**) $c/a$=0.35  (**f**) $c/a$=0.40

(**g**) $c/a$=0.45

**Figure 10.** The buckling behavior of the sections with different $c/a$.

According to the calculation formula for elastic local buckling of channel section considering plate interaction proposed by Shafer [11], the formula for critical stress of elastic local buckling of simple lipped equal limb angle steel section is as follows:

$$\sigma_{crl} = \frac{k_l \pi^2 E}{12(1-v^2)}(t/a)^2 \tag{1}$$

The relationship between $k_l$ and $c/a$ is obtained by regression fitting, as shown in Figure 11. It can be seen that the relationship between $k_l$ and $c/a$ does not change as the ratio of length to thickness ($a/t$) varies.

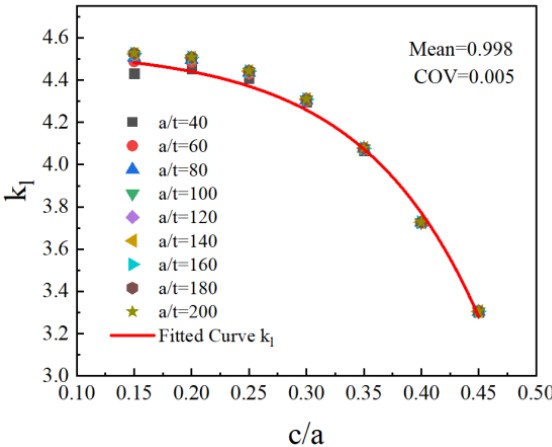

**Figure 11.** Relationship between $k_l$ and $c/a$.

Based on the above analysis, the suggested expression of $k_l$ is as follows:

$$k_l = 8.41(c/a) - 17.23(c/a)^2 + 3.26 \qquad (2)$$

To verify the proposed formula, the critical stresses of elastic local buckling $\sigma_{crl}$ of 12 different sections with different parameters were calculated by CUFSM, and the results were compared with the results $\sigma_{crl2}$ by Equation (2), as listed in Table 3.

**Table 3.** Comparison of critical stress for elastic local buckling of simple lipped equal limb angle steel member.

| $a/t$ | $c/a$ | $\sigma_{crl}$ /MPa | $\sigma_{crl2}$ /MPa | $\frac{\sigma_{crl}}{\sigma_{crl2}}$ |
|---|---|---|---|---|
| 50 | 0.25 | 332.81 | 336.06 | 0.99 |
| | 0.35 | 325.26 | 320.94 | 1.01 |
| | 0.45 | 287.79 | 278.80 | 1.03 |
| 70 | 0.25 | 170.77 | 171.46 | 1.00 |
| | 0.35 | 166.33 | 163.74 | 1.02 |
| | 0.45 | 146.83 | 142.24 | 1.03 |
| 90 | 0.25 | 103.55 | 103.72 | 1.00 |
| | 0.35 | 100.66 | 99.06 | 1.02 |
| | 0.45 | 88.90 | 86.05 | 1.03 |
| 110 | 0.25 | 69.39 | 69.43 | 1.00 |
| | 0.35 | 67.40 | 66.31 | 1.02 |
| | 0.45 | 59.50 | 57.60 | 1.03 |
| | Mean | | | 1.015 |
| | COV | | | 0.013 |

According to the comparison in Table 3, the mean value of the ratio between the CUFSM algorithm and the proposed formula in this paper is 1.015, and the coefficient of variation (COV) is 0.013. The comparison verified the accuracy of the proposed formula.

Then, by changing the ratio of length to thickness ($a/t$) and the ratio of the primary lip to long limb ($c/a$), 315 simple lipped unequal limb angle sections were selected to investigate the influence coefficient $k_f$ on the critical stress of elastic local buckling.

The critical stresses of elastic local buckling of the 315 simple lipped unequal limb angle sections are shown in Table 4. It can be seen that the influence of the limb length ratio decreases with the increase of the length-to-thickness ratio.

**Table 4.** Critical stresses of elastic local buckling for simple lipped unequal limb angle member (units: MPa).

| $a/t$ | $a/b$ | Critical Stresses of Local Buckling with Different $c/a$ | | | | | | |
|---|---|---|---|---|---|---|---|---|
| | | 0.2 | 0.25 | 0.3 | 0.35 | 0.4 | 0.45 | 0.5 |
| 40 | 1.1 | 552.72 | 554.64 | 547.00 | 529.72 | 497.39 | 448.58 | 393.56 |
| | 1.2 | 572.17 | 573.67 | 564.86 | 544.61 | 509.69 | 455.90 | 398.27 |
| | 1.3 | 583.39 | 584.61 | 574.92 | 553.01 | 515.77 | 459.98 | 400.96 |
| | 1.4 | 589.87 | 590.38 | 581.53 | 558.48 | 519.58 | 462.62 | 402.72 |
| | 1.5 | 594.07 | 594.48 | 586.23 | 562.44 | 522.30 | 464.51 | 403.92 |
| 60 | 1.1 | 249.15 | 248.54 | 244.47 | 236.18 | 221.23 | 199.60 | 175.08 |
| | 1.2 | 257.26 | 256.51 | 252.37 | 242.83 | 226.48 | 203.04 | 177.11 |
| | 1.3 | 261.69 | 260.80 | 256.45 | 246.60 | 229.44 | 204.92 | 178.18 |
| | 1.4 | 264.60 | 263.60 | 259.03 | 249.06 | 231.36 | 206.03 | 178.87 |
| | 1.5 | 266.72 | 265.65 | 260.90 | 250.85 | 232.62 | 206.83 | 179.37 |
| 80 | 1.1 | 140.65 | 140.13 | 137.75 | 132.99 | 124.55 | 112.33 | 98.50 |
| | 1.2 | 145.25 | 144.56 | 142.07 | 136.74 | 127.44 | 114.18 | 99.60 |
| | 1.3 | 147.87 | 147.08 | 144.38 | 138.87 | 129.06 | 115.21 | 100.22 |
| | 1.4 | 149.59 | 148.73 | 145.89 | 140.26 | 130.12 | 115.88 | 100.63 |
| | 1.5 | 150.71 | 149.94 | 146.99 | 141.27 | 130.88 | 116.37 | 100.92 |
| 100 | 1.1 | 90.16 | 89.74 | 88.23 | 85.16 | 79.75 | 71.89 | 63.03 |
| | 1.2 | 93.17 | 92.64 | 90.96 | 87.56 | 81.59 | 73.10 | 63.75 |
| | 1.3 | 94.79 | 94.27 | 92.47 | 88.92 | 82.62 | 73.77 | 64.15 |
| | 1.4 | 95.81 | 95.25 | 93.46 | 89.82 | 83.28 | 74.19 | 64.42 |
| | 1.5 | 96.56 | 95.97 | 94.19 | 90.42 | 83.76 | 74.49 | 64.61 |
| 120 | 1.1 | 62.68 | 62.35 | 61.30 | 59.15 | 55.37 | 49.94 | 43.77 |
| | 1.2 | 64.75 | 64.38 | 63.19 | 60.82 | 56.68 | 50.76 | 44.27 |
| | 1.3 | 65.88 | 65.47 | 64.25 | 61.77 | 57.39 | 51.22 | 44.56 |
| | 1.4 | 66.62 | 66.18 | 64.94 | 62.38 | 57.84 | 51.52 | 44.75 |
| | 1.5 | 67.16 | 66.70 | 65.41 | 62.80 | 58.17 | 51.74 | 44.88 |
| 140 | 1.1 | 46.04 | 45.89 | 45.26 | 43.91 | 41.40 | 37.46 | 32.80 |
| | 1.2 | 47.57 | 47.35 | 46.58 | 44.99 | 42.09 | 37.79 | 32.95 |
| | 1.3 | 48.43 | 48.15 | 47.31 | 45.60 | 42.49 | 37.99 | 33.04 |
| | 1.4 | 48.95 | 48.68 | 47.78 | 45.99 | 42.76 | 38.14 | 33.12 |
| | 1.5 | 49.33 | 49.04 | 48.13 | 46.28 | 42.97 | 38.25 | 33.18 |
| 160 | 1.1 | 35.27 | 35.15 | 34.66 | 33.62 | 31.69 | 28.67 | 25.11 |
| | 1.2 | 36.44 | 36.26 | 35.67 | 34.45 | 32.23 | 28.93 | 25.23 |
| | 1.3 | 37.08 | 36.88 | 36.23 | 34.91 | 32.54 | 29.09 | 25.30 |
| | 1.4 | 37.49 | 37.27 | 36.59 | 35.22 | 32.74 | 29.20 | 25.36 |
| | 1.5 | 37.79 | 37.55 | 36.86 | 35.44 | 32.90 | 29.29 | 25.40 |
| 180 | 1.1 | 27.87 | 27.78 | 27.39 | 26.57 | 25.04 | 22.66 | 19.84 |
| | 1.2 | 28.80 | 28.66 | 28.19 | 27.22 | 25.47 | 22.86 | 19.93 |
| | 1.3 | 29.31 | 29.14 | 28.63 | 27.59 | 25.71 | 22.99 | 19.99 |
| | 1.4 | 29.64 | 29.45 | 28.92 | 27.83 | 25.87 | 23.07 | 20.04 |
| | 1.5 | 29.87 | 29.68 | 29.13 | 28.01 | 26.00 | 23.14 | 20.07 |
| 200 | 1.1 | 22.58 | 22.50 | 22.18 | 21.52 | 20.29 | 18.35 | 16.07 |
| | 1.2 | 23.33 | 23.21 | 22.83 | 22.05 | 20.63 | 18.52 | 16.15 |
| | 1.3 | 23.75 | 23.61 | 23.19 | 22.35 | 20.83 | 18.62 | 16.19 |
| | 1.4 | 24.01 | 23.86 | 23.42 | 22.54 | 20.96 | 18.69 | 16.23 |
| | 1.5 | 24.20 | 24.04 | 23.59 | 22.69 | 21.06 | 18.74 | 16.26 |

To obtain the influence of limb length ratio on the critical stresses of elastic local buckling of simple lipped unequal limb angle steel section, according to Equation (1), it can be changed to

$$\sigma_{crl} = \frac{k_l k_f \pi^2 E}{12(1 - v^2)} (t/a)^2 \tag{3}$$

According to the results of Table 4 and Equation (3), the influence coefficient $k_f$ of limb length can be expressed as a nonlinear relation of $a/b$:

$$k_f = 0.292 - 1.06(a/b)^2 + 0.339(a/b) \tag{4}$$

To verify the proposed formula, the critical stresses of elastic local buckling $\sigma_{crl}$ of 10 different sections with different parameters were calculated by CUFSM, and the results were compared with the results $\sigma_{crl2}$ by Equation (4), as listed in Table 5.

**Table 5.** Comparison of critical stress for elastic local buckling of simple lipped unequal limb angle steel member.

| $a/t$ | $a/b$ | $\sigma_{crl}$ /MPa | $\sigma_{crl2}$ /MPa | $\frac{\sigma_{crl}}{\sigma_{crl2}}$ |
|---|---|---|---|---|
| | 1.1 | 175.34 | 171.57 | 1.02 |
| | 1.2 | 179.67 | 176.16 | 1.02 |
| 70 | 1.3 | 182.14 | 179.64 | 1.01 |
| | 1.4 | 183.80 | 182.01 | 1.01 |
| | 1.5 | 185.02 | 183.27 | 1.01 |
| | 1.1 | 71.10 | 69.48 | 1.02 |
| | 1.2 | 72.85 | 71.34 | 1.02 |
| 110 | 1.3 | 73.86 | 72.75 | 1.02 |
| | 1.4 | 74.49 | 73.71 | 1.01 |
| | 1.5 | 74.95 | 74.22 | 1.01 |
| | Mean | | | 1.016 |
| | COV | | | 0.005 |

According to the comparison in Table 5, the mean value of the ratio between the CUFSM algorithm and the proposed formula is 1.016, and the coefficient of variation (COV) is 0.005. The comparison verified the accuracy of the proposed formula.

*3.3. Critical Stress of Elastic Local Buckling of Complex Lipped Angle Members*

In this section, the 153 equal limb sections with complex lipped edges were selected to explore the influence of complex edges on the critical stresses of elastic local buckling. The critical stresses of elastic local buckling for the complex lipped sections are listed in Table 6.

It can be seen from Table 6 that when $c/a$ increases, the critical stresses of elastic local buckling of the section are almost unchanged. Meanwhile, it can be seen from Table 6 that when $d/c$ increases, the critical stresses of elastic local buckling increase first and then decrease. Based on the above analysis, it can be concluded that the critical stress of elastic local buckling of the section is less affected by the edge size for the complex edge section. However, the critical stress of elastic local buckling of the angle steel section can be significantly improved by the complex edge.

In conclusion, the complex edge has a certain enhancement effect on the critical stress of elastic local buckling of the section. It is suggested that the influence coefficient $k_l$ of angle section with a complex edge can be taken as 4.74 fitting from Table 6. The formula for critical stress of elastic local buckling of complex lipped equal limb angle steel section is as follows:

$$\sigma_{crl} = \frac{k_l \pi^2 E}{12(1 - v^2)} (t/a)^2 \quad k_l = 4.74 \tag{5}$$

**Table 6.** Critical stresses of elastic local buckling for complex lipped equal limb angle member (units: MPa).

| $a/t$ | $d/c$ | Critical Stress of Elastic Local Buckling with Different $c/a$ | | | | | |
|---|---|---|---|---|---|---|---|
| | | **0.20** | **0.25** | **0.30** | **0.35** | **0.40** | **0.45** |
| 40 | 0.50 | 553.17 | 578.69 | 583.60 | 583.57 | 581.68 | 579.08 |
| | 0.75 | 576.74 | 587.51 | 588.06 | 585.83 | 582.64 | 579.00 |
| | 1.00 | 586.61 | 591.19 | 589.58 | 585.87 | 580.17 | - |
| 60 | 0.50 | 259.53 | 263.49 | 263.18 | 261.95 | 260.57 | 258.98 |
| | 0.75 | 265.15 | 265.33 | 263.92 | 262.27 | 260.59 | 256.00 |
| | 1.00 | 266.99 | 265.92 | 264.05 | 261.97 | 259.00 | - |
| 80 | 0.50 | 148.84 | 149.18 | 148.43 | 147.47 | 146.51 | 145.59 |
| | 0.75 | 150.55 | 149.86 | 148.73 | 147.57 | 146.46 | 144.00 |
| | 1.00 | 151.22 | 150.085 | 148.73 | 147.32 | 145.60 | - |
| 100 | 0.50 | 96.12 | 95.82 | 95.18 | 94.52 | 93.90 | 93.32 |
| | 0.75 | 96.79 | 96.08 | 95.29 | 94.54 | 93.85 | 92.30 |
| | 1.00 | 97.04 | 96.15 | 95.25 | 94.37 | 93.26 | - |
| 120 | 0.50 | 67.07 | 66.71 | 66.18 | 65.67 | 65.21 | 64.77 |
| | 0.75 | 67.41 | 66.83 | 66.24 | 65.68 | 65.16 | 64.10 |
| | 1 | 67.52 | 66.86 | 66.21 | 65.54 | 64.74 | - |
| 140 | 0.5 | 49.39 | 49.04 | 48.63 | 48.26 | 47.93 | 47.62 |
| | 0.75 | 49.59 | 49.12 | 48.66 | 48.26 | 47.93 | 47.10 |
| | 1.00 | 49.66 | 49.13 | 48.64 | 48.16 | 47.58 | - |
| 160 | 0.50 | 37.88 | 37.58 | 37.26 | 36.97 | 36.70 | 36.45 |
| | 0.75 | 38.00 | 37.62 | 37.28 | 36.96 | 36.66 | 36.10 |
| | 1.00 | 38.04 | 37.63 | 37.25 | 36.88 | 36.42 | - |
| 180 | 0.50 | 29.98 | 29.71 | 29.44 | 29.21 | 29.00 | 28.81 |
| | 0.75 | 30.05 | 29.74 | 29.45 | 29.20 | 28.97 | 28.50 |
| | 1.00 | 30.07 | 29.74 | 29.43 | 29.14 | 28.78 | - |
| 200 | 0.50 | 24.30 | 24.07 | 23.85 | 23.67 | 23.49 | 23.33 |
| | 0.75 | 24.35 | 24.09 | 23.86 | 23.66 | 23.47 | 23.10 |
| | 1.00 | 24.37 | 24.09 | 23.84 | 23.61 | 23.31 | - |

To verify the proposed formula, the critical stresses of elastic local buckling $\sigma_{crl}$ of 12 different sections with different parameters were calculated by CUFSM, and the results were compared with the results $\sigma_{crl2}$ by Equation (5), as listed in Table 7.

According to the comparison in Table 7, the mean value of the ratio between the CUFSM algorithm and the proposed formula is 1.020, and the coefficient of variation (COV) is 0.013. The comparison verified the accuracy of the proposed formula.

The limb length influence coefficient $k_f$ of the simple lipped angle steel section is also applicable to the complex lipped angle steel section. To verify the formula, according to the section size in Table 1, 315 complex lipped unequal limb angle steel sections ($d/c = 0.5$) were selected. The critical elastic local buckling stress of the section was obtained by CUFSM.

It can be seen from Table 8 that the complex lipped angle steel section is similar to that of the simple lipped angle steel section. The ratio of length to width has little effect on the critical stress of complex lipped angle steel with the increase of the length to width ratio of the section. When the width-to-thickness ratio of the limb is large, the length ratio of the limb has little influence on the complex edge section, but it has great influence on the unequal limb section with the small width-to-thickness ratio of the limb.

**Table 7.** Comparison of critical stress for elastic local buckling of complex lipped equal limb angle steel member.

| $a/t$ | $c/a$ | $d/c$ | $\sigma_{crl}$ /MPa | $\sigma_{crl2}$ /MPa | $\frac{\sigma_{crl}}{\sigma_{crl2}}$ |
|---|---|---|---|---|---|
| | 0.25 | | 375.57 | 371.69 | 1.01 |
| 50 | 0.35 | 0.50 | 375.78 | 371.69 | 1.01 |
| | 0.45 | | 371.18 | 371.69 | 1.00 |
| | 0.25 | | 194.24 | 189.64 | 1.02 |
| 70 | 0.35 | 0.50 | 192.42 | 189.64 | 1.01 |
| | 0.45 | | 190.15 | 189.64 | 1.00 |
| | | 0.50 | 118.23 | 114.72 | 1.03 |
| 90 | 0.25 | 0.75 | 118.58 | 114.72 | 1.03 |
| | | 1.00 | 118.69 | 114.72 | 1.03 |
| | | 0.50 | 79.28 | 76.79 | 1.03 |
| 110 | 0.25 | 0.75 | 79.47 | 76.79 | 1.03 |
| | | 1.00 | 79.52 | 76.79 | 1.04 |
| | Mean | | | | 1.020 |
| | COV | | | | 0.013 |

According to the calculation results in Table 8 and Equation (4), the limb length influence coefficient $k_f$ can be expressed as a nonlinear relationship of limb length ratio $a/b$, and the formula for critical stress of elastic local buckling of complex lipped unequal limb angle steel section is as follows:

$$\sigma_{crl} = \frac{k_l k_f \pi^2 E}{12(1-v^2)}(t/a)^2 \tag{6}$$

$$k_l = 4.74 \tag{7}$$

$$k_f = 0.292 - 1.06(a/b)^2 + 0.339(a/b) \tag{8}$$

To verify the proposed formula, the critical stresses of elastic local buckling $\sigma_{crl}$ of 10 different sections with different parameters were calculated by CUFSM, and the results were compared with the results $\sigma_{crl2}$ by Equations (6)–(8), as listed in Table 9.

According to the comparison in Table 9, the mean value of the ratio between the CUFSM algorithm and the proposed Equations (6)–(8) is 1.033, and the coefficient of variation (COV) is 0.014. The comparison verified the accuracy of the proposed formula.

To conclude, the suggested calculation formulas of critical stress of elastic local buckling of lipped angle steel section are as follows:

$$\sigma_{crl} = \frac{k_l k_f \pi^2 E}{12(1-v^2)}(t/a)^2 \tag{9}$$

$$k_l = \begin{cases} 4.74 \\ 8.41(c/a) - 17.23(c/a)^2 + 3.26 \end{cases} \tag{10}$$

$$k_f = \begin{cases} 1 \\ 0.292 - 1.060(a/b)^2 + 0.339(a/b) \end{cases} \tag{11}$$

**Table 8.** Critical stresses of elastic local buckling for complex lipped unequal limb angle member (units: MPa).

| $a/t$ | $a/b$ | Critical Stress of Elastic Local Buckling with Different $c/a$ | | | | | | |
|---|---|---|---|---|---|---|---|---|
| | | 0.20 | 0.25 | 0.30 | 0.35 | 0.40 | 0.45 | 0.50 |
| 40 | 1.1 | 594.77 | 618.50 | 624.26 | 624.28 | 622.13 | 619.10 | 615.67 |
| | 1.2 | 616.28 | 637.90 | 644.06 | 644.10 | 641.80 | 638.56 | 634.85 |
| | 1.3 | 627.99 | 648.71 | 655.10 | 655.16 | 652.77 | 649.41 | 645.54 |
| | 1.4 | 634.56 | 655.74 | 662.30 | 661.66 | 659.32 | 656.48 | 652.52 |
| | 1.5 | 639.31 | 660.85 | 666.77 | 665.42 | 663.06 | 660.29 | 657.32 |
| 60 | 1.1 | 278.62 | 281.26 | 280.69 | 279.29 | 277.70 | 276.10 | 274.49 |
| | 1.2 | 286.99 | 289.62 | 289.02 | 287.54 | 285.85 | 284.14 | 282.41 |
| | 1.3 | 291.52 | 294.24 | 293.62 | 292.09 | 290.35 | 288.58 | 286.78 |
| | 1.4 | 294.47 | 297.24 | 296.62 | 295.06 | 293.29 | 291.47 | 289.63 |
| | 1.5 | 296.61 | 299.44 | 298.81 | 297.23 | 295.43 | 293.58 | 291.70 |
| 80 | 1.1 | 159.45 | 159.45 | 158.60 | 157.61 | 156.64 | 155.73 | 154.68 |
| | 1.2 | 164.03 | 164.02 | 163.12 | 162.67 | 161.05 | 160.07 | 159.11 |
| | 1.3 | 166.54 | 166.17 | 165.61 | 164.53 | 163.47 | 162.46 | 161.46 |
| | 1.4 | 168.17 | 168.17 | 167.22 | 166.12 | 165.05 | 164.01 | 162.99 |
| | 1.5 | 169.36 | 169.36 | 168.40 | 167.29 | 166.19 | 165.14 | 164.10 |
| 100 | 1.1 | 102.79 | 102.45 | 101.73 | 100.97 | 100.26 | 99.59 | 98.95 |
| | 1.2 | 105.76 | 105.31 | 104.59 | 103.87 | 103.21 | 102.58 | 101.92 |
| | 1.3 | 107.34 | 106.88 | 106.14 | 105.40 | 104.71 | 104.07 | 103.44 |
| | 1.4 | 108.36 | 107.90 | 107.15 | 106.39 | 105.69 | 105.03 | 104.39 |
| | 1.5 | 109.11 | 108.65 | 107.90 | 107.12 | 106.40 | 105.73 | 105.08 |
| 120 | 1.1 | 71.66 | 71.22 | 70.66 | 70.13 | 69.64 | 69.19 | 68.76 |
| | 1.2 | 73.74 | 73.27 | 72.72 | 72.21 | 71.69 | 71.21 | 70.75 |
| | 1.3 | 74.82 | 74.34 | 73.78 | 73.25 | 72.77 | 72.33 | 71.85 |
| | 1.4 | 75.53 | 75.04 | 74.46 | 73.93 | 73.44 | 72.98 | 72.54 |
| | 1.5 | 76.04 | 75.55 | 74.96 | 74.42 | 73.92 | 73.46 | 73.01 |
| 140 | 1.1 | 52.77 | 52.37 | 51.94 | 51.55 | 51.19 | 50.86 | 50.52 |
| | 1.2 | 54.31 | 53.90 | 53.46 | 53.04 | 52.67 | 52.32 | 51.99 |
| | 1.3 | 55.10 | 54.67 | 54.24 | 53.85 | 53.48 | 53.12 | 52.78 |
| | 1.4 | 55.00 | 55.18 | 54.74 | 54.34 | 53.98 | 53.64 | 53.29 |
| | 1.5 | 55.97 | 55.55 | 55.10 | 54.69 | 54.33 | 53.99 | 53.66 |
| 160 | 1.1 | 40.47 | 40.13 | 39.79 | 39.48 | 39.19 | 38.93 | 38.68 |
| | 1.2 | 41.65 | 41.29 | 40.93 | 40.61 | 40.33 | 40.70 | 39.82 |
| | 1.3 | 42.25 | 41.89 | 41.55 | 41.23 | 40.94 | 40.67 | 40.41 |
| | 1.4 | 42.63 | 42.27 | 41.92 | 41.62 | 41.33 | 41.06 | 40.79 |
| | 1.5 | 42.92 | 42.55 | 42.20 | 41.89 | 41.61 | 41.34 | 41.07 |
| 180 | 1.1 | 32.01 | 31.73 | 31.44 | 31.19 | 30.97 | 30.76 | 30.56 |
| | 1.2 | 32.93 | 32.63 | 32.30 | 32.10 | 31.88 | 31.66 | 31.46 |
| | 1.3 | 33.41 | 33.12 | 32.83 | 32.58 | 32.35 | 32.14 | 31.94 |
| | 1.4 | 33.72 | 33.42 | 33.14 | 32.89 | 32.66 | 32.44 | 32.24 |
| | 1.5 | 33.94 | 33.63 | 33.35 | 33.11 | 32.88 | 32.66 | 32.45 |
| 200 | 1.1 | 25.96 | 25.70 | 25.47 | 25.30 | 25.09 | 24.92 | 24.76 |
| | 1.2 | 26.69 | 26.44 | 26.21 | 26.01 | 25.82 | 25.65 | 25.48 |
| | 1.3 | 26.87 | 26.83 | 26.60 | 26.39 | 26.21 | 26.04 | 25.87 |
| | 1.4 | 27.33 | 27.08 | 26.85 | 26.64 | 26.45 | 26.28 | 26.12 |
| | 1.5 | 27.51 | 27.25 | 27.02 | 26.82 | 26.63 | 26.46 | 26.29 |

**Table 9.** Comparison of critical stress for elastic local buckling of complex lipped unequal limb angle steel member.

| *a/t* | *a/b* | $\sigma_{crl}$ /MPa | $\sigma_{crl2}$ /MPa | $\frac{\sigma_{crl}}{\sigma_{crl2}}$ |
|---|---|---|---|---|
| 70 | 1.1 | 205.67 | 198.70 | 1.04 |
| | 1.2 | 211.80 | 204.02 | 1.04 |
| | 1.3 | 214.79 | 208.05 | 1.03 |
| | 1.4 | 216.72 | 210.79 | 1.03 |
| | 1.5 | 218.13 | 212.25 | 1.03 |
| 110 | 1.1 | 83.46 | 80.47 | 1.04 |
| | 1.2 | 85.86 | 82.62 | 1.04 |
| | 1.3 | 87.18 | 84.25 | 1.03 |
| | 1.4 | 88.03 | 85.36 | 1.03 |
| | 1.5 | 88.58 | 85.95 | 1.03 |
| | Mean | | | 1.033 |
| | COV | | | 0.014 |

## 4. Critical Stress of the Elastic Distortional Buckling of Simple Lipped Angle Columns

In this section, the CUFSM software was used to analyze the distortional buckling model of simple lipped angle columns with different sizes. Young [22] and Zhang [23] carried out tests on simple lipped equal limb angle steel and sections without edges, respectively, and the tests showed that no distortional buckling occurred for all members. According to a large number of analyses by CUFSM, it was also found that there is no distortional buckling for the angle steel section. Meanwhile, there was no obvious distortional buckling deformation for the simple lipped angle steel with different sizes.

Finally, the CUFSM software was used to analyze several axial compression members with different sizes of simple lipped unequal limb angle steel sections, as shown in Figure 12. The results showed that there was still no distortional buckling mode. Similarly, Young [24] did not find distortional buckling in analyzing a simple lipped unequal limb angle steel section. Therefore, the effect of distortional buckling can be ignored for the simple lipped angle steel section.

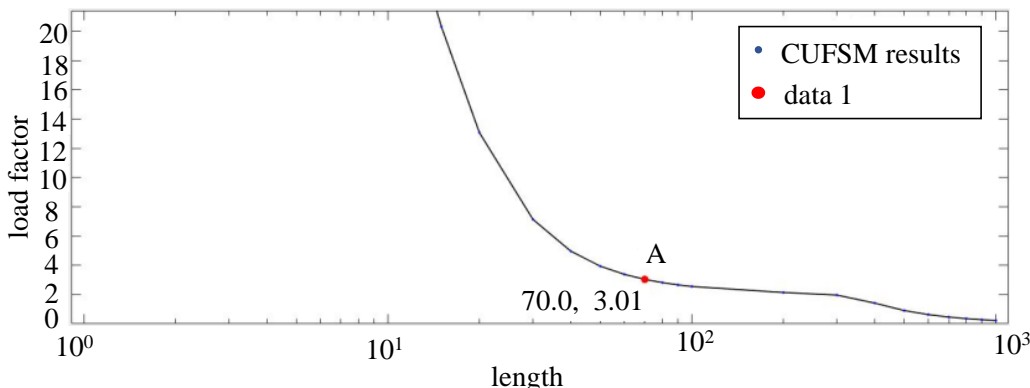

**Figure 12.** CUFSM analysis curve of simple lipped unequal angle steel.

## 5. Critical Stress of Elastic Distortional Buckling of Complex Lipped Angle Members

*5.1. Section without Distortional Point*

The above analysis indicated that the complex lipped angle steel sections showed distortional buckling. However, not all the sections had an obvious second minimum point, as shown in Figure 13.

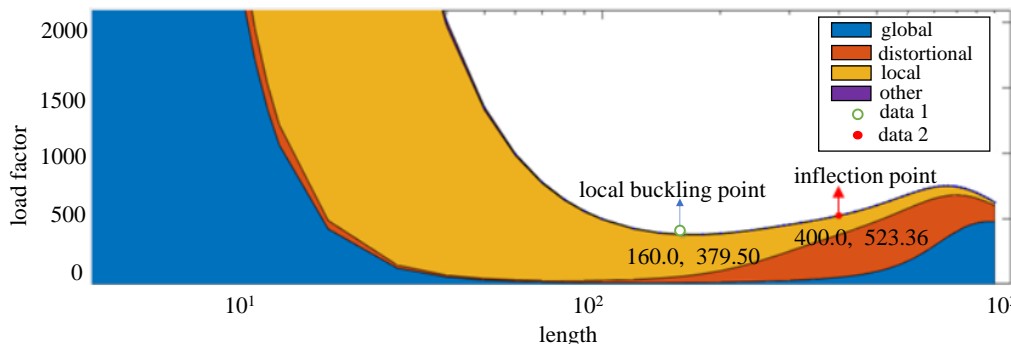

**Figure 13.** CUFSM analysis result of the section without distortional point.

From the numerical analysis, it indicated that the section could be divided into two categories by the size of the first edge: (1) when $c \leq 0.3a$, the curve doesn't have the second minimum point, and the cross-section belongs to the section without a distortional point; (2) when $c > 0.3a$, the second minimum point appears on the curve, and the cross-section belongs to the section with distortional point. The critical stress of elastic distortional buckling of some complex lipped angle steel sections is listed in Table 10.

**Table 10.** Critical stresses of elastic local buckling for complex lipped angle member (units: MPa).

| *a* | *b* | *c* | *d* | *a/b* | *d/c* | *c/a* | $\sigma_{crd}$ | Series |
|---|---|---|---|---|---|---|---|---|
|     | 160 | 32 | 16 | 1.0 | 0.50 |     | 217 |  |
|     | 145 | 32 | 24 | 1.1 | 0.75 |     | 302 |  |
|     | 133 | 32 | 32 | 1.2 | 1.00 |     | 350 |  |
| 160 | 123 | 32 | 16 | 1.3 | 0.50 | 0.2 | 261 | without distortional point |
|     | 115 | 32 | 24 | 1.4 | 0.75 |     | 338 |  |
|     | 105 | 32 | 32 | 1.5 | 1.00 |     | 375 |  |
|     | 160 | 64 | 32 | 1.0 | 0.50 |     | 445 |  |
|     | 145 | 64 | 48 | 1.1 | 0.75 |     | 490 |  |
|     | 133 | 64 | 64 | 1.2 | 1.00 |     | 438 |  |
| 160 | 123 | 64 | 32 | 1.3 | 0.50 | 0.4 | 504 | with distortional point |
|     | 115 | 64 | 48 | 1.4 | 0.75 |     | 520 |  |
|     | 105 | 64 | 64 | 1.5 | 1.00 |     | 450 |  |
|     | 240 | 48 | 24 | 1.0 | 0.50 |     | 130 |  |
|     | 218 | 48 | 36 | 1.1 | 0.75 |     | 190 |  |
|     | 200 | 48 | 48 | 1.2 | 1.00 |     | 220 |  |
| 240 | 185 | 48 | 24 | 1.3 | 0.50 | 0.2 | 158 | without distortional point |
|     | 170 | 48 | 36 | 1.4 | 0.75 |     | 210 |  |
|     | 160 | 48 | 48 | 1.5 | 1.00 |     | 240 |  |
|     | 240 | 96 | 48 | 1.0 | 0.50 |     | 290 |  |
|     | 218 | 96 | 72 | 1.1 | 0.75 |     | 320 |  |
|     | 200 | 96 | 96 | 1.2 | 1.00 |     | 300 |  |
| 240 | 185 | 96 | 48 | 1.3 | 0.50 | 0.4 | 327 | with distortional point |
|     | 170 | 96 | 72 | 1.4 | 0.75 |     | 340 |  |
|     | 160 | 96 | 96 | 1.5 | 1.00 |     | 304 |  |

Although CUFSM results for a simply supported cross-section showed no distortional point (second minimum point) appeared in the section, significant distortional buckling deformation can occur in the section, and distortional buckling can be seen from the existing test results [25] of complex lipped unequal angle steels.

The main methods for solving elastic distortional buckling are the methods proposed by Hancock [19] and Schafer [11]. In this paper, both are used to estimate the critical stresses of elastic distortional buckling. The method proposed by Hancock has been introduced into the Chinese code [6]. However, for the section without a distortional point ($c \leq 0.3a$),

there is a strong correlation between distortional buckling and local buckling as well as global buckling, as shown in Figure 13. The deformation of distortional buckling is the largest at the wavelength of 400 mm, and the proportion of local buckling, distortional buckling, and global buckling is 27.4%, 63%, and 9.6%, respectively. It indicated that the error between the critical stresses of elastic distortional buckling and the calculation result of CUFSM was obvious for the section without distortional point by the method proposed by Hancock. Table 11 lists the analysis results of the complex equal limb angle steel and unequal limb angle steel section. The mean value of the ratio between the calculated results of CUFSM and the Hancock method is 0.73, and the coefficient of variation (COV) is 0.14. Therefore, the analytical method cannot accurately predict the critical stress of distortional buckling for the section without a distortional point.

**Table 11.** Comparison of CUFSM results between analytical method and numerical analysis method.

| *Ca-b-c-d-t* | *a/b* | Hancock | CUFSM | $\frac{\sigma_{crC}}{\sigma_{crH}}$ |
|---|---|---|---|---|
| | | $\sigma_{crH}$(MPa) | $\sigma_{crC}$(MPa) | |
| C240-240-60-30-2 | 1.0 | 240.50 | 181.30 | 0.75 |
| C240-218-60-30-2 | 1.1 | 240.50 | 200.27 | 0.83 |
| C240-200-60-30-2 | 1.2 | 240.50 | 211.88 | 0.88 |
| C240-185-60-30-2 | 1.3 | 240.50 | 200.69 | 0.83 |
| C240-171-60-30-2 | 1.4 | 240.50 | 226.92 | 0.94 |
| C240-160-60-30-2 | 1.5 | 240.50 | 229.73 | 0.96 |
| C280-280-70-35-2 | 1.0 | 241.16 | 151.55 | 0.63 |
| C280-254-70-35-2 | 1.1 | 241.16 | 167.90 | 0.70 |
| C280-233-70-35-2 | 1.2 | 241.16 | 179.23 | 0.74 |
| C280-215-70-35-2 | 1.3 | 241.16 | 185.62 | 0.77 |
| C280-200-70-35-2 | 1.4 | 241.16 | 190.96 | 0.79 |
| C280-185-70-35-2 | 1.5 | 241.16 | 193.70 | 0.80 |
| C360-360-90-45-2 | 1.0 | 236.94 | 114.76 | 0.48 |
| C360-327-90-45-2 | 1.1 | 236.94 | 127.01 | 0.54 |
| C360-300-90-45-2 | 1.2 | 236.94 | 135.14 | 0.57 |
| C360-276-90-45-2 | 1.3 | 236.94 | 141.00 | 0.60 |
| C360-257-90-45-2 | 1.4 | 236.94 | 144.51 | 0.61 |
| C360-240-90-45-2 | 1.5 | 236.94 | 147.62 | 0.62 |
| Mean | | | | 0.73 |
| COV | | | | 0.14 |

For the section without a distortional point, the critical stress at the distortional buckling position of the section can be set as the critical stress value of elastic distortional buckling of the section [21]. However, the critical stress of the elastic distortional buckling is too conservative. The result is not unique when the stress at the distortional buckling deformation position is directly taken as the section. In this paper, the critical stresses of elastic distortional buckling are taken as the lesser of the elastic critical stresses corresponding to the maximum deformation of distortion buckling (inflection point of curve) and the maximum probability of distortional buckling.

As shown in Figure 14, by analyzing point D corresponding to the minimum stress in the distortional buckling curve, point C (inflection point of curve) of the maximum section distortional buckling deformation is indirectly obtained. It is necessary to perform the probability analysis to find the maximum probability position of the distortional buckling.

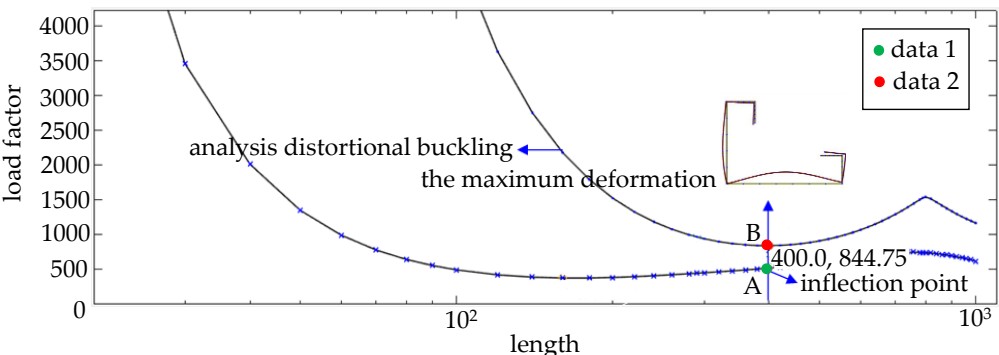

**Figure 14.** The maximum deformation of distortion buckling.

*5.2. Section with Distortional Point*

The critical stresses of elastic distortional buckling of the section with distortional point can be calculated by the method proposed by Hancock [19]. In this section, the distortional buckling of channel steel with flange is compared with that of the section with distortional buckling, as shown in Figure 15.

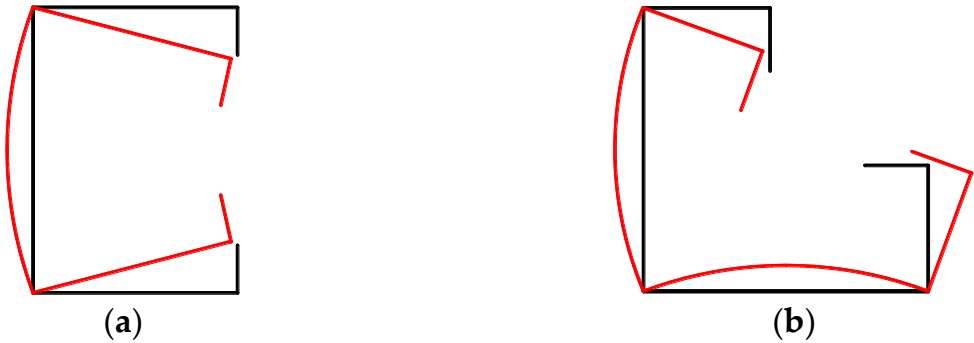

**Figure 15.** Distortional buckling of angle member: (**a**) Channel section with flange; (**b**) Complex lipped equal limb angle steel ($c > 0.3a$).

The distortional buckling deformation mechanism of the channel steel section is the combination of flange and edge, as shown in Figure 16. Similar to the distortional buckling deformation of channel steel, the edge combination is taken as the distortional buckling deformation of the complex lipped angle steel section.

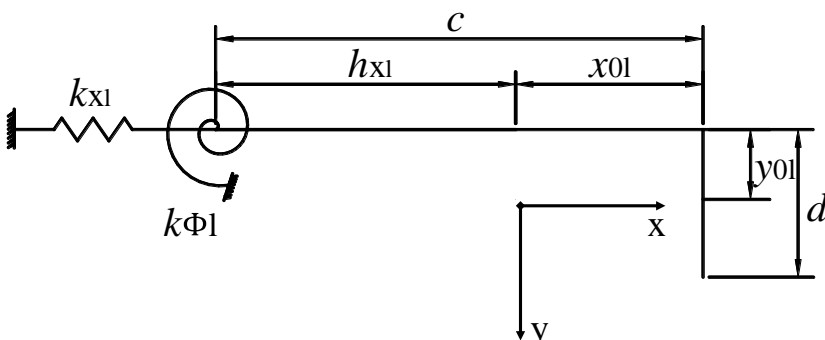

**Figure 16.** Simplified calculation model for distortional buckling.

The coordinate axis in Figure 16 is the centroid coordinate axis of the distortional buckling model. The coordinates of the shear center are ($x_{01}, y_{01}$),

$$x_{01} = c^2/2(c+d) \tag{12}$$

$$h_{y1} = y_{01} = -d^2/2(c+d) \tag{13}$$

$$x_{01} - h_{x1} = c \tag{14}$$

where $c$ is the length of the primary edge, $d$ is the length of the secondary edge, $h_{x1}$ is the distance between the support to the centroid and can be determined as follows:

$$h_{x1} = -\left(c^2 + 2cd\right)/2(c+d) \tag{15}$$

Concerning the lateral restraint of the limb, the lateral restraint value $k_{x1}$ is taken as 0. Therefore, the critical stress of elastic distortional buckling of the section with distortional point can be calculated by Equation (16).

$$\sigma_{crd} = \frac{E}{2A_1}\left[\alpha_1 + \alpha_2 - \sqrt{(\alpha_1 + \alpha_2)^2 - 4\alpha_3}\right] \tag{16}$$

$$\alpha_1 = \frac{\eta}{\beta_1}(\beta_2 + 0.039I_{t1}\lambda^2) + \frac{k\varphi_1}{\beta_1\eta E} \tag{17}$$

$$\alpha_2 = \eta\left(I_{y1} - 2y_{01}\frac{\beta_3}{\beta_1}\right) \tag{18}$$

$$\alpha_3 = \eta\left(\alpha_1 I_{y1} - \frac{\eta}{\beta_1}\beta_3^2\right) \tag{19}$$

$$\beta_1 = h_{x1}^2 + (I_{x1} + I_{y1})/A_1 \tag{20}$$

$$\beta_2 = I_{w1} + I_{x1}(x_{01} - h_{x1})^2 \tag{21}$$

$$\beta_3 = I_{xy1}(x_{01} - h_{x1}) = I_{xy1}c \tag{22}$$

$$\beta_4 = \beta_2 + (y_{01} - h_{y1})\left[I_{y1}(y_{01} - h_{y1})\right] - 2\beta_3 \tag{23}$$

$$\lambda = 4.80\left(\frac{I_{x1}c^2a}{t^3}\right)^{0.25} \tag{24}$$

$$\eta = (\pi/\lambda)^2 \tag{25}$$

where $E$ is the elastic modulus, $A_1$ is the sectional area, $\lambda$ is the buckling half-wavelength, $c$ is the length of the primary edge, $k_\varphi$ is the rotational stiffness, and the product of inertia $I_{xy}$ is used in these equations because the x and y axes are not principal axes.

The critical stress of elastic distortional buckling for sections with distortional points is calculated by CUFSM software and the Hancock method, respectively. The calculation results are shown in Table 12.

The comparison in Table 12 includes the complex lipped angle section with equal or unequal limbs. It was found that the calculation results of CUFSM were in good agreement with the calculation results of the Hancock method, and the difference between the two methods was basically within 10%. The mean ratio between CUFSM and that of the Hancock method is 1.033, and the coefficient of variation (COV) is 0.048.

Conclusively, for the section without distortional points, the critical stresses of elastic distortional buckling are taken as the lesser of the elastic critical stresses corresponding to the maximum deformation of distortion buckling (inflection point of curve) and the maximum probability of distortional buckling. For the section with distortional points, both the method proposed by Hancock and CUFSM can be used to determine the critical stress.

**Table 12.** Comparison of Hancock and CUFSM results for sections with distortional points.

| Ca-b-c-d-t | a/b | Hancock | CUFSM | $\frac{\sigma_{crC}}{\sigma_{crH}}$ |
| | | $\sigma_{crH}$(MPa) | $\sigma_{crC}$(MPa) | |
|---|---|---|---|---|
| C240-240-108-54-2 | 1.0 | 321.54 | 302.39 | 0.94 |
| C240-218-108-54-2 | 1.1 | 321.54 | 307.88 | 0.96 |
| C240-200-108-54-2 | 1.2 | 321.54 | 317.82 | 0.99 |
| C240-185-108-54-2 | 1.3 | 321.54 | 327.78 | 1.02 |
| C240-171-108-54-2 | 1.4 | 321.54 | 334.34 | 1.04 |
| C240-160-108-54-2 | 1.5 | 321.54 | 338.77 | 1.05 |
| C280-280-126-63-2 | 1.0 | 268.80 | 257.41 | 0.96 |
| C280-254-126-63-2 | 1.1 | 268.80 | 270.61 | 1.01 |
| C280-233-126-63-2 | 1.2 | 268.80 | 279.14 | 1.04 |
| C280-215-126-63-2 | 1.3 | 268.80 | 284.77 | 1.05 |
| C280-200-126-63-2 | 1.4 | 268.80 | 288.58 | 1.07 |
| C280-185-126-63-2 | 1.5 | 268.80 | 291.10 | 1.08 |
| C360-360-162-81-2 | 1.0 | 200.66 | 198.4 | 0.99 |
| C360-327-162-81-2 | 1.1 | 200.66 | 208.64 | 1.04 |
| C360-300-162-81-2 | 1.2 | 200.66 | 215.26 | 1.07 |
| C360-276-162-81-2 | 1.3 | 200.66 | 219.65 | 1.09 |
| C360-257-162-81-2 | 1.4 | 200.66 | 221.00 | 1.10 |
| C360-240-162-81-2 | 1.5 | 200.66 | 224.00 | 1.11 |
| Mean | | | | 1.033 |
| COV | | | | 0.048 |

## 6. Conclusions

In this paper, a series of numerical and theoretical researches on cold-formed steel members with complex edges were carried out. Finite strip method software CUFSM was used to calculate the elastic critical stress of four kinds of cross-section. The cross-sectional deformation diagram and the elastic critical stress were obtained. Then, the elastic critical stresses of the sections were determined by the analytical method and numerical method, respectively, to propose a method to determine the critical stress of the complex edge section. The following conclusions can be drawn based on the present research work.

(1) To calculate the critical stresses of elastic local buckling of the angle steel section, the interaction between the panels should be considered. The binding effect of different edge forms on the limb is not the same. The constraint effect of a complex edge on the limb is always greater than that of a simple edge.

(2) The critical stress of elastic local buckling increases with the increase of the limb length ratio for both the simple lipped unequal limb angle steel and complex lipped unequal limb angle steel. Formulas for calculating the critical stress of elastic local buckling of lipped angle steel section considering the restraint between plates are proposed and verified.

(3) According to the different methods of determining the critical stresses of elastic distortional buckling, the complex lipped angle steel sections are divided into two categories by judging the size of the first edge: (1) when $c \leq 0.3a$, the curve doesn't have the second minimum point, and the cross-section belongs to the section without distortional point; (2) when $c > 0.3a$, the second minimum point appears on the curve, and the cross-section belongs to the section with distortional point.

(4) For the section without a distortional point, the critical stresses of elastic distortional buckling are taken as the lesser of the elastic critical stresses corresponding to the maximum deformation of distortion buckling (inflection point of curve) and the maximum probability of distortional buckling. For the section with a distortional point, the critical stress of the second minimum point (distortional point) in the curve can be taken as the critical stresses elastic of distortional buckling of the complex

lipped angle section. The method proposed by Hancock can be used to calculate the elastic critical stresses of distortional buckling for this kind of section.

However, the calculation formula of critical stress of distortional buckling for the section without distortional point was not given in this paper. It is necessary to conduct follow-up research.

**Author Contributions:** Conceptualization, J.Z. and S.P.; software, S.P.; validation, S.P.; formal analysis, S.P.; writing—original draft preparation, B.L.; supervision, A.L.; investigation, A.L.; resources, A.L.; data curation, A.L. All authors have read and agreed to the published version of the manuscript.

**Funding:** The research presented in this paper was financially supported by the Natural Science Foundation of China (Grant No. 51908511), Key Research and Promotion Project (Scientific and Technological Project) of Henan Province (Grant Nos. 212102310283 and 212102310286), the Key Research Projects of Henan Higher Education Institutions (Grant NO. 20A560001) and the China Postdoctoral Science Foundation (Grant NO. 2019M662532).

**Institutional Review Board Statement:** Not applicable.

**Informed Consent Statement:** Not applicable.

**Data Availability Statement:** Not applicable.

**Conflicts of Interest:** The authors declared that they have no conflict of interest in this work. We declare we do not have any commercial or associative interest representing a conflict of interest in connection with the manuscript entitled, "Critical stress determination of local and distortional buckling of lipped angle columns under axial compression" by J.Z., B.L., A.L., S.P.

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
