# Peer review of "Critical Stress Determination of Local and Distortional Buckling of Lipped Angle Columns under Axial Compression"

_buildings, doi:10.3390/buildings12060712_

Round 1

Reviewer 1 Report

I am favorable of the publication of this article.

Author Response

Point 1: I am favorable of the publication of this article.

Response 1: Thank you for your comments.

Reviewer 2 Report

This paper investigates the stability bearing capacity of angle sections with complex edges under axial compression by the software CUFSM. The research goals include (1) proposing a calculation formula for critical stress of elastic local buckling (low slenderness ratio) of complex lipped angle (CLA) sections by regression analysis (a total of 1296 different CLA sections were analyzed by CUFSM), (2) Analyzing the critical stress of elastic distortional buckling (high slenderness ratio) of complex lipped angle sections (a total of 918 CLA sections without/with distortional point), and (3) Verifying the accuracy of Hancock method for calculating the critical stress of elastic distortional buckling of CLA sections with distortional point. There is only one problem the author is considering for this paper. The simulation results from CUFSM are similar to Hancock/Schafer’s (the developer of CUFSM) method. Have the authors done the relevant research about the algorithm of CUFSM? If it is possible that the development of CUFSM uses Hancock/Schafer’s method as a reference? If so, the simulation results and conclusions made in this research need to be reconsidered. The authors make some contributions to the revision of Equation (1) shown in this research based on a great number of simulation results. The reviewer agrees that this research can provide useful guidelines for the design of cold-formed steel angle columns. The reviewer recommends the authors can clarify the following issues:

  1. Page 1, line 14 in the Abstract. Please add a reference for the CUFSM (the reviewer used this software before) the first time it appears. The authors can cite the software’s official link from John Hopkins University. Also, please write the version number (CUFSM 4 or CUFSM 5) for this software.
  2. Page 2, Figure 2, please ensure the font size in this Figure is uniform. Please explain the x0, y0, x, and y-axis stands for what. Please also explain what the limb (a and b in the Figure) is and what the lip (c and d shown in this Figure) is here (the first time a new term appears), not in Section 3.1.
  3. Page 2, line 69, add a reference for the North American Specification code (AISI S100 or S240 or S400). Please also specify which version (2012 or 2016).
  4. Page 3, line 100. Please explain why the CLA sections without distortional points are not verified in this research. Is that due to the description shown in lines 349 and 350 that the analytical method cannot accurately predict the critical stress of distortional buckling for the section without a distortional point?
  5. Page 4, Figure 4, please make sure the font size and font type are uniform in this Figure. Also, due to CUFSM is developed by MATLAB. Please save the analysis result to the PNG format directly other than using a screenshot. In this way, the readers will not see the gray background in Figures 4 and 5.
  6. Page 5, line 153, please add a reference for the Q345 steel. It comes from which code (China or European)?
  7. Page 14, Figure 9, please explain the read point’s meaning (data 1) and save the analysis result to the PNG format.
  8. Page 14, Figure 10. Please explain the meaning of data 1 to data 5. And is there any difference between data 1 and data 3? Because data 1 and data 3 seem to use the same marker in this Figure.
  9. Page 17, Figure 11, this Figure lack legend. Please add it. Please also make sure the font size and font type shown in one Figure are uniform.

Finally, the reviewer will give two suggestions for the authors. (1) Please ensure that all the font types and sizes in one Figure are unified. (2) Please check the minor grammar and typos in this manuscript.

Author Response

Point 1: The simulation results from CUFSM are similar to Hancock/Schafer’s (the developer of CUFSM) method. Have the authors done the relevant research about the algorithm of CUFSM? If it is possible that the development of CUFSM uses Hancock/Schafer’s method as a reference? If so, the simulation results and conclusions made in this research need to be reconsidered.

Response 1: Thank you for your comment. The CUFSM developed by Hancock/Schafer has been widely adopted for the analysis of cold-formed steel sections. Consequently, the algorithm of CUFSM was not emphasized in this paper. However, it should be mentioned that the analytical method proposed by Hancock/Schafer is not suitable for the complex lipped angle section in this paper. The analytical method cannot accurately obtain the critical stress of elastic local buckling for a section consisting of multiple plates. Therefore, the CUFSM was used to conduct the numerical analysis to fit a new formula to determine the critical stress of elastic local buckling for the complex lipped angle members in this paper.

Point 2: Page 1, line 14 in the Abstract. Please add a reference for the CUFSM (the reviewer used this software before) the first time it appears. The authors can cite the software’s official link from John Hopkins University. Also, please write the version number (CUFSM 4 or CUFSM 5) for this software.

Response 2: Thank you for your careful comment. The software’s official link (www.ce.jhu.edu/bschafer/cufsm) has been added in the reference and the version number CUFSM 5 has been added in the revised manuscript. However, we politely think that it is not commonplace to use reference in the Abstract. The official link has been added at the end of the introduction, where the software appears the first time in the main boby of the paper.

Point 3: Page 2, Figure 2, please ensure the font size in this Figure is uniform. Please explain the x0, y0, x, and y-axis stands for what. Please also explain what the limb (a and b in the Figure) is and what the lip (c and d shown in this Figure) is here (the first time a new term appears), not in Section 3.1.

Response 3: Thank you for your comment. Figure 2 has been revised to ensure that the font size is uniform. The meanings of the symbols have been explained. The x0 and y0 are the centroidal axes, and the x and y are the centroidal principal axes of the section. a is the length of the long limb and b is the length of the short limb. c and d are the length of the primary lip and the secondary lip, respectively.

Point 4: Page 2, line 69, add a reference for the North American Specification code (AISI S100 or S240 or S400). Please also specify which version (2012 or 2016).

Response 4: Thanks for your careful comment. AISI S100 (2016) was used in this paper and the reference has been added.

Point 5: Page 3, line 100. Please explain why the CLA sections without distortional points are not verified in this research. Is that due to the description shown in lines 349 and 350 that the analytical method cannot accurately predict the critical stress of distortional buckling for the section without a distortional point?

Response 5: Thanks for your comment. We agree with the reviewer that the analytical method cannot accurately predict the critical stress of distortional buckling for the section without a distortional point. According to the numerical analysis in this paper, for the section without distortional point (c≤0.3a), there is a strong correlation between distortional buckling and local buckling as well as global buckling. The deformation of distortional buckling is the largest at the wavelength of 400 mm, and the proportion of local buckling, distortional buckling and global buckling is 27.4%, 63% and 9.6%, respectively. It indicated that the error between the critical stresses of elastic distortional buckling and the calculation result of CUFSM was obvious for the section without distortional point by the method proposed by Hancock. Table 11 listed the analysis results of the complex equal limb angle steel and unequal limb angle steel section. The mean value of the ratio between the calculated results of CUFSM and Hancock method is 0.73, and the coefficient of variation (COV) is 0.14. Therefore, the analytical method cannot accurately predict the critical stress of distortional buckling for the section without distortional point.

Point 6: Page 4, Figure 4, please make sure the font size and font type are uniform in this Figure. Also, due to CUFSM is developed by MATLAB. Please save the analysis result to the PNG format directly other than using a screenshot. In this way, the readers will not see the gray background in Figures 4 and 5.

Response 6: Thank you for your careful comment. Figure 4 and Figure 5 have been revised according to your comment.

Point 7: Page 5, line 153, please add a reference for the Q345 steel. It comes from which code (China or European)?

Response 7: Thank you for your question. Q345 steel comes from Chinese code Technical specification for cold-formed thin-walled steel structures (GB 50018-2020). The reference has been added.

Point 8: Page 14, Figure 9, please explain the read point’s meaning (data 1) and save the analysis result to the PNG format.

Response 8: Thanks for your comment. Figure 9 (Figure 12 in the revised manuscript) has been revised.

The red point is an ordinary point and this figure is used to illustrate that there is no distortional buckling mode in the analysis of simple lipped unequal limb angle steel section. Data 1 means that when the length is 70mm, the load factor Pcr is 3.01Mpa.

Point 9: Page 14, Figure 10. Please explain the meaning of data 1 to data 5. And is there any difference between data 1 and data 3? Because data 1 and data 3 seem to use the same marker in this Figure. 

Response 9: Thanks for your insightful comment. Figure 10 has been revised in the manuscript and the key points have been marked carefully. Date 1 means the local buckling point and date 2 means the inflection point.

Point 10: Page 17, Figure 11, this Figure lack legend. Please add it. Please also make sure the font size and font type shown in one Figure are uniform.

Response 10: Thank you for your comment. Figure 11 has been revised. The legend has been added and the font has been revised according to the comment.

Reviewer 3 Report

The paper presented a numerical investigation using finite elements software concerning the local buckling of the cold-formed steel columns under a pure axial loading scenario. The paper is interesting since the case of study and the finding are well presented and discussed. However, the following comments are required to be considered by the authors to improve the article’s presentation further:

  • Abstract: the main difference between this study from the previous studies in this field needs to be highlighted clearly.
  • The abstract could be more informative by providing results; some results at the conclusion/verifications/comparisons need to be highlighted clearly.
  • The limitations of this study need to be highlighted and discussed in the introduction.
  • Section 2: For better understanding by the readers of this paper, the authors must provide more details about the verification study of the suggested FE modeling concept, such as more photos from the FE analysis and comparisons of the results that can confirm the validity of modeling suggestion.
  • Section 3: support your discussions with some photos related to the buckling behavior and/or stresses obtained from the FE analyses of the studied models.
  • The current argument of conclusions is good but could be further improved by mentioning your study limits and suggesting some future research topics.

Author Response

Point 1: Abstract: the main difference between this study from the previous studies in this field needs to be highlighted clearly.

Response 1: Thank you for your insightful comment. The main difference between this study from the previous studies is the type of the cross section. The critical stress of local and distortional buckling of complex lipped angle section were investigated in this paper. The abstract has been revised according to your comment.

Point 2: The abstract could be more informative by providing results; some results at the conclusion/verifications/comparisons need to be highlighted clearly.

Response 2: Thank you for your useful comment. We have added some analysis results in the abstract to make it more informative.

Point 3: The limitations of this study need to be highlighted and discussed in the introduction.

Response 3: Thanks for your comment. This paper focused on the critical stress of lipped angle columns under axial compression. For the columns under combined bending and axial compression, further research is needed. The limitation of this study has been added at the end of the introduction.

Point 4: Section 2: For better understanding by the readers of this paper, the authors must provide more details about the verification study of the suggested FE modeling concept, such as more photos from the FE analysis and comparisons of the results that can confirm the validity of modeling suggestion.

Response 4: Thanks for the comment. The example in Direct Strength Method (DSM) Design Guide 2006 was used to verify the FE modeling concept in this paper.

The section size and the buckling behaviors of the example are shown in the following figure (Figure 5 and figure 6 in the revised manuscript).

From comparison of diagram and critical stress between the design guide and the CUFSM in this paper, it can be seen that the analysis results of CUFSM are basically the same as those of the design guide. In this way, the accuracy of the CUFSM modeling can be validated.

Point 5: Section 3: support your discussions with some photos related to the buckling behavior and/or stresses obtained from the FE analyses of the studied models.

Response 5: Thanks for your comment. The buckling behavior of the section obtained from the FE analyses has been added in the revised manuscript.

When a/t=40, the buckling behavior are showed in the following Figure (Figure 10 in the revised manuscript):

Point 6: The current argument of conclusions is good but could be further improved by mentioning your study limits and suggesting some future research topics.

Response 6: Thank you for your constructive comment. The limitation and future research topics have been added at the end of the conclusions.
